# Inhomogeneity in electronic phase and flat band in magnetic kagome metal $Co_3Sn_2S_2$
Sandy Adhitia Ekahana [1], Satoshi Okamoto [2], Jan Dreiser [1], Loïc Roduit[1], Igor Plokhikh [1], Dariusz Jakub Gawryluk [1], Andrew Hunter[3], Anna Tamai [3] & Yona Soh [1] ✉

$Co_3Sn_2S_2$ has been reported to be a Weyl semimetal with $c$-axis ferromagnetism below a Curie temperature of 177 K. Despite the large interest in $Co_3Sn_2S_2$, the magnetic structure is still unclear. Recent studies have challenged the magnetic phase diagram of $Co_3Sn_2S_2$ by reporting unusual magnetic phases including the presence of exchange bias. Here we show, using X-ray Magnetic Circular Dichroism, a shift in the magnetization hysteresis loop, reminiscent of exchange bias and establish that the magnetic moment in Co arises from the spin, with negligible orbital moment. At 6 K, using spatially-resolved angle-resolved photoemission spectroscopy, we detect a butterfly-shaped electronic band structure at small regions of the sample distinct from the known ferromagnetic band structure. Our density functional theory calculations suggest that the butterfly bands correspond to an antiferromagnetic phase. Separately, we detect a sharp flat band at the Fermi level at some regions in the sample, which we attribute to a surface state. These different electronic states found in a stoichiometric intermetallic invite further efforts to explore the origin and nature of the electronic inhomogeneity associated to magnetism on the mesoscale.

Magnetic frustration frequently manifests in materials with a kagome lattice structure where the inherent shared triangle configuration poses a challenge to satisfy certain magnetic interactions to minimize the total energy, giving rise to an antiferromagnetic (AFM) fluctuation[1], spin fluctuation[2], and other frustrated magnet situations[3–6]. Meanwhile, a kagome ferromagnet (FM) $Fe_3Sn_2$ with a unidirectional magnetic moment that undergoes spin reorientation from the $c$-axis towards the kagome plane at a lower temperature[7] settles comfortably at a certain minimum energy state without suffering magnetic frustration. $Co_3Sn_2S_2$, which is also reported to be a $c$-axis FM[8,9] with $T_C \cong 177$ K and a saturated magnetic moment of 0.3 $\mu_B$ per Co[10,11], almost has that privilege if not due to several reports that contradict the simple FM phase paradigm.

There are conflicting reports regarding the magnetic phase diagram of $Co_3Sn_2S_2$ although the searches of Weyl points were conducted[12–15] assuming such a FM phase. The sample in one of the first reports of magnetic anisotropy[9] shows a hysteresis loop typical for a FM (square and symmetric shape, no exchange bias) at low temperature with no report of an unusual magnetic phase. However, it was reported in 2017 that $Co_3Sn_2S_2$ harbors an anomalous magnetic phase just below the Curie temperature ($T_C \sim 175$ K) and above a temperature called $T_A \approx 125$ K (or in the temperature range 125–175 K) before it adopts a fully FM phase at lower temperatures with magnetic moments along the $c$-axis as long as the applied external magnetic field is small[16]. Increasing this external magnetic field makes this anomalous phase disappear and shows a pure FM phase.

This unique phase was revisited through muon spin rotation ($\mu$SR) experiments and reported to be a competition between the FM phase and an in-plane AFM phase, whose phase diagram depends on the doping concentration of indium in $Co_3Sn_{2-x}In_xS_2$, with increasing In promoting a mixed phase of FM and AFM[17,18]. On the other hand, neutron scattering experiments on $Co_3Sn_{2-x}In_xS_2$ concluded that instead of a phase coexistence of pure FM and AFM phases, the system is a homogeneous phase starting from a pure $c$-axis FM evolving to a canted FM with the canting angle from the $c$-axis increasing from 0° to 65° as the indium doping is varied from 0 to 30%[19,20]. Further density functional theory (DFT) calculations suggest that the cobalt magnetic moment may be canted by a small angle of 1.5° off the $c$-axis at 128 K[21], whereas a study based on pair distribution functions on powder suggests a local instability distortion (random canting) of the Co moment to be ~ 20° off the perpendicular axis[22] while maintaining an average out-of-plane moment in the long range order. Recent studies with non-linear optics report a homogeneous but temperature-dependent magnetic phase, with a $c$-axis FM at $T_A < T < T_c$ and a canted FM with small canting angle below $T_A$[23]. Therefore, it is still a debated question whether an AFM phase coexists with the FM phase at $T_A < T < T_c$ and whether the magnetic configuration in $Co_3Sn_2S_2$ at $T < T_A$ is a pure $c$-axis out-of-kagome-plane FM or a canted FM.

[1]Paul Scherrer Institute, Villigen, Switzerland. [2]Materials Science and Technology Division, Oak Ridge National Laboratory, Oak Ridge, TN, USA. [3]Department of Quantum Matter Physics, University of Geneva, Geneva, Switzerland. ✉e-mail: yona.soh@psi.ch

We would like to stress the difficulty of detecting a minority AFM phase in the sea of a FM majority phase when the propagation vector, **q**, of both considered magnetic maximal symmetry Shubnikov supgroups is equal to [0, 0, 0]. Due to the "hidden" nature of the AFM order not exhibiting any net magnetic moment, when AFM is a small minority phase in the presence of a FM majority phase, techniques such as magnetometry fail since the signal is dominated by the FM phase. When an AFM order is characterized by a propagation vector other than [0, 0, 0], in addition to nuclear Bragg peaks associated with the crystal, new purely magnetic reflections appear in the magnetically ordered state enabling easier detection of the AFM order if the magnetic moment is sufficiently large. However, when **q** = [0, 0, 0] and the magnetic moment is small, which is the case of $Co_3Sn_2S_2$, detecting a small amount of a minority AFM phase is challenging since there are no new reflections and the locations of the AFM reflections coincide with those of the nuclear Bragg and FM peaks, which have much larger scattering intensities.

In this paper, we show the existence of electronic inhomogeneity at $T = 6\,K$ by using micro-focus angle-resolved photoemission spectroscopy ($\mu$-ARPES) with 6.01 eV photon energy ($4^{th}$ harmonic continuous laser), where a butterfly-shaped electronic phase is detected in an otherwise majority electronic phase corresponding to the FM phase. We also find out that the orbital moment in Co is negligible based on X-Ray Magnetic Circular Dichroism (XMCD). We demonstrate that the butterfly-shaped electronic band structure is of magnetic origin by cycling the temperature up to the paramagnetic (PM) phase, where it disappears, and show that it reappears as we cool the system to $T = 6\,K$. Our discovery opens a new perspective that the phase below $T_A$ is not entirely FM and a minority phase in $Co_3Sn_2S_2$ exists at a low temperature of 6 K. These findings enrich our understanding of phase coexistence as we spatially display the coexistence with direct visualization of the band structure owing to the advanced laser $\mu$-ARPES technique. Ultimately, this finding invites further exploration of the spin and band structure analysis of such topological kagome systems and opens paths for future engineering where the coexistence of different electronic and magnetic phases can be controlled and utilized in electronic and magnetic applications such as in spintronics and exchange bias.

## Results and discussions

### Crystal structure and magnetic properties of $Co_3Sn_2S_2$

The $Co_3Sn_2S_2$ compound crystallizes in Shandite-like structure of $Ni_3Pb_2S_2$ archetype[24,25] with space group R-3m, number 166, and lattice constant $a = 5.38$ Å and $c = 13.19$ Å in a conventional unit cell with hexagonal

cross section. The composition of $Co_3Sn_2S_2$ in the conventional unit cell consists of three kagome layers translated from each other (Fig. 1a). Each kagome layer comprises Co atoms that lie on the kagome lattice site with one Sn atom lying at the center of the Co kagome lattice or star of David shape. Meanwhile, each triangle forming the kagome lattice, which consists of Co atoms, hosts a S atom hovering above the triangle center of mass alternating on the top side (S1) and the bottom side (S2) (which have equivalent Wyckoff position (6c)) of the kagome lattice (green prisms in Fig. 1) as we circle around the triangles forming the star of David. On the same triangles, we have Sn atoms also hovering on the triangle center of mass projection but with longer bond lengths (blue prisms) than the S-kagome bond (green prisms) and placed opposite to the S atoms. The Sn atoms are shared between neighboring kagome layers on the top and on the bottom, making this $Co_3Sn_2S_2$ a quasi-layered structure due to these shared Sn atomic layers. From this structure, $Co_3Sn_2S_2$ allows two different cleaving planes: cleaving between the Sn-S layer or cleaving between the S-kagome layer. These two cleaving planes are not equivalent as the formation energy is smaller in breaking the Sn-S bond than the S-kagome bond[12]. Therefore, it is rare to observe the kagome termination or the S termination whose layers underneath are the Sn and S layer stacked consecutively (see Supplementary Fig. 1 (iii) and (iv)). We can also deduce from a simple observation that the bond length between the Co atom on the kagome plane and the S atom on the nearby S1 or S2 layer is the shortest (2.20 Å) (green prism vertices connecting S and Co atoms). Therefore, it is more probable to have the S termination with the kagome layer underneath (S1) or the Sn termination.

For the magnetic structure, only the Co atoms give rise to the magnetic moment, where DFT calculations have concluded that the minimum energy configuration is a FM phase (R-3m' magnetic space group No 166.101), where all Co moments are pointing out of the kagome plane (c-axis FM or $(0, 0, m_z)$)[12,26]. Our synchrotron-based ARPES result (with beam spot of ~50 $\mu$m x 50 $\mu$m) in Fig. 1b also captures the FM bands, which agree with our DFT calculation ($\mu_{Co} = 0.35\,\mu_B$) if we introduce a "band renormalization" to simulate the correlation effect and raise the calculated band minimum closer to the Fermi level ($E_F$) to match the ARPES data, as also performed in other publications[15,27]. We do this by dividing the energy scale on the DFT result by a certain factor (1.43 in this case) and thus raising the band energy position. Our synchrotron-based ARPES result agrees qualitatively with other published results[13,28] showing the general consensus of FM in this material. Realizing a pure AFM phase is not favorable energetically as both the "chiral" (R-3m space group No 166.97) and the "in/out" magnetic configuration (also being realized in R-3m' space group by two

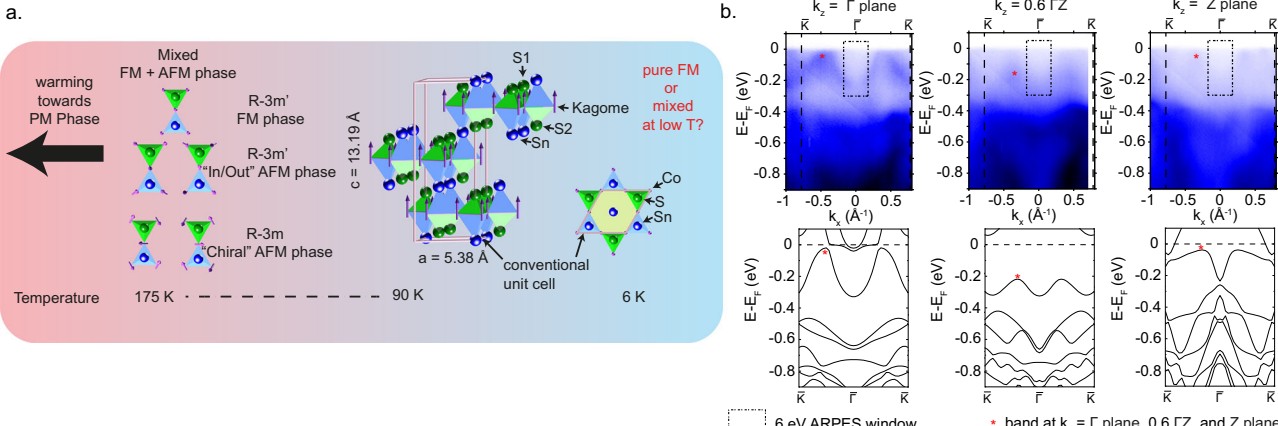

**Fig. 1 | $Co_3Sn_2S_2$ crystal, magnetic structure, and $k_z$ dependent ARPES result. a** The crystal structure shows the kagome layers of Co-Sn with S and Sn atoms alternatingly occupying the center of the kagome triangle at a certain distance from the kagome layer creating a quasi-layer of S and a quasi-layer of Sn, with the latter being shared between the adjacent kagome layers. The schematic shows the common understanding that $Co_3Sn_2S_2$ is a paramagnet (PM) above the transition

temperature $T_C \sim 177\,K$, below which there is an open question on whether the magnetic phase is a pure c-axis ferromagnetic (FM) phase, a canted FM phase, or a mixture of FM and antiferromagnetic (AFM) phases. The AFM phase can exist in two possible configurations called "In/Out" and "Chiral" AFM phase. **b** Synchrotron based $k_z$ dependent ARPES result showing general agreement with the c-axis ferromagnetic band structure calculation assuming $\mu_{Co} = 0.35\,\mu_B$.

**Fig. 2 | Schematic of XMCD experiment and results. a** Schematic of XAS measurement where the x-ray and the *B* field are collinear and antiparallel to each other. The XAS result of the Co $L_{2,3}$ peak measured using total electron yield from both circular polarizations show a clear dichroism. **b** Temperature dependent sum-rule analysis of the XMCD Co peak showing that the orbital component is negligible and the magnetism is dominated by the spin-effective component, which vanishes around 175 K in accordance with the magnetic phase transition. **c** Angle dependent sum-rule analysis shows a negligible orbital component while the value of the effective spin drops to ∼$\frac{2}{3}$ at 60° x-ray incident angle relative to normal incidence, suggesting the system has a strong magnetic anisotropy and largely maintains the moment along the *c*-axis at rotating fields of 6.8 T despite the coercive field for magnetization along the *c*-axis being one order of magnitude smaller. **d** The hysteresis curve at 0° and 60° x-ray incident angles at *T* = 50 K obtained by tracing the normalized XMCD $L_3$ peak shows the signal saturation and a relatively rectangular shape with a visible exchange bias. The saturated signal shows a reduction to ∼$\frac{2}{3}$ at 60° x-ray incident angle compared to normal incidence indicating that the magnetic moment does not fully follow the magnetic field direction. The exchange bias suggests the possible existence of an AFM phase.

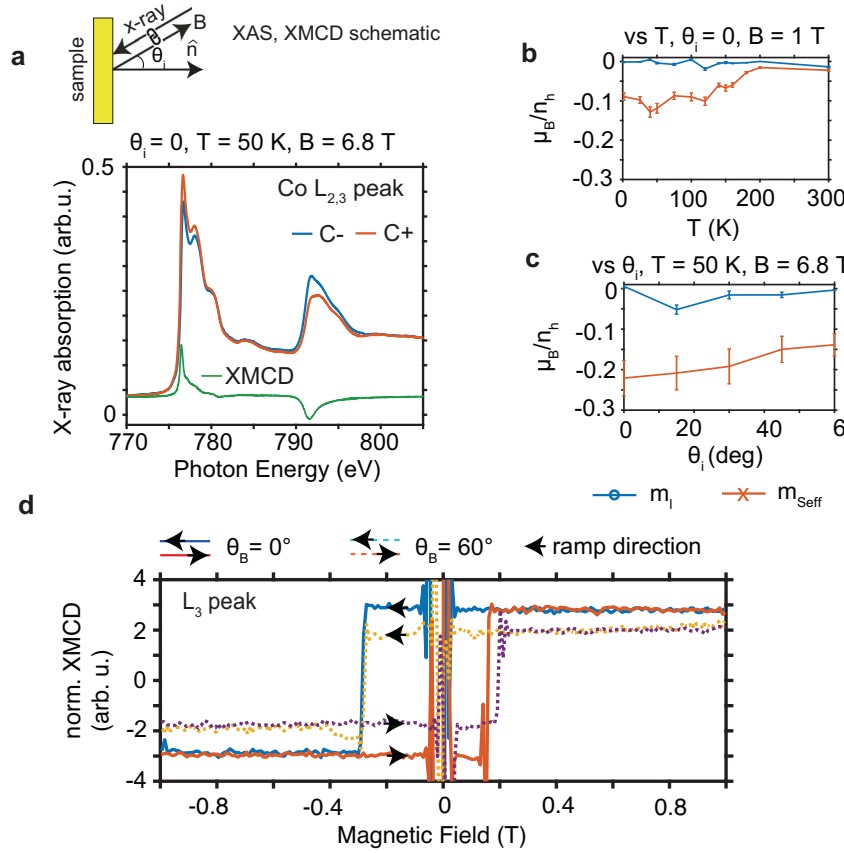

spin components of magnetic cobalt in ($m_x$, $2m_x$, $m_z$) configuration) (see Fig. 1a) require a constrained calculation, with the R-3m AFM phase reported to be lower in energy by ∼1 meV than the R-3m'AFM phase $\left(E_{FM} < E_{AFMR-3m} < E_{AFMR-3m'}\right)$[26]. Our DFT calculation shows that the FM phase is lower in energy by ~50 meV than both AFM phases.

## X-ray magnetic circular dichroism (XMCD)

To explore the magnetic phase in $Co_3Sn_2S_2$ further, we measure the element-specific magnetization at the cobalt $L_{2,3}$ edges with XMCD in the total electron yield (TEY) mode, performed at the X-Treme beam line at the Paul Scherrer Institute[29]. The advantage of studying the magnetic behavior using XMCD is that it can disentangle the orbital and spin magnetic moment contributions separately. The schematic of the XMCD experiment setup is given in Fig. 2a showing that the photon **k** vector is coupled to the direction of the applied magnetic field (a positive magnetic field is opposite to the direction of the photon **k** vector and vice-versa), and the sample normal direction can be rotated for angle-dependent measurements. With this measurement, we explore how the x-ray absorption spectrum (XAS) at the cobalt $L_{2,3}$ edges varies with left and right circularly polarized light (C- and C + ) (Fig. 2a), from which we can calculate the XMCD signal from both $L_3$ and $L_2$ edges. The sample was freshly cleaved in situ prior to the measurement at *T* = 50 K and at a pressure of *p*~$10^{-11}$ mbar for the results presented in Fig. 2a, c and d. Meanwhile, the data shown in Fig. 2b was obtained on a carbon-capped sample after being freshly cleaved in another ultra-high vacuum (UHV) chamber and transferred in an atmospheric environment to the X-Treme chamber. This explains the quantitatively different sum-rule results we present below, where the carbon-capped sample shows a relatively smaller effective spin value than the freshly cleaved sample. However, the results from both samples agree qualitatively.

Our XAS result agrees with the published XANES data[30], where the focus was on the electronic structure but not on magnetism. We can see in

Fig. 2a that the XAS signals from C+ and C- show a visible difference, from which the XMCD = $I_{C+} - I_{C-}$ can be calculated; the integrated XMCD signal is proportional to the cobalt magnetization projected along the x-ray wavevector. We performed the sum-rule analysis (detailed explanation in Methods) on the XMCD data to obtain the orbital ($m_l$) and effective spin magnetic moment ($m_{S_{eff}}$) as a function of temperature (Fig. 2b) at normal incidence and also as a function of the incidence angle (Fig. 2c) at *T* = 50 K. Error bars in Fig. 2b, c represent the uncertainty predominantly due to the background subtraction. We can see in Fig. 2b that the orbital magnetic moment is relatively small at all temperatures while the effective spin moment vanishes at *T*~175 K, in agreement with the magnetic transition towards the PM phase. The relatively negligible orbital moment implies that the magnetic moment in $Co_3Sn_2S_2$ comes mainly from the spin. This result is at odds with the reported large negative orbital magnetism (~-$3\mu_B$) inferred from the field dependence of the peak in the density of states (DOS) observed around $E_F$ by scanning tunneling spectroscopy (STS)[31]. We highlight here that XMCD is a well-established element-specific magnetic probe, and it provides a direct measurement of the orbital and effective spin moments, in contrast to the STS.

The angle-dependent result shows a decreasing value for the effective spin projected onto the x-ray direction at higher incidence angles with $m_{S_{eff},60°} \approx \frac{2}{3}m_{S_{eff},0°}$ (Fig. 2c), showing a strongly anisotropic behavior agreeing with the preferred *c*-axis FM orientation. However, since $m_{S_{eff},60°}$ is not strictly $\frac{1}{2}m_{S_{eff},0°}$, it is an indication that either the spin is a little tilted from the surface normal, as a result of responding to the angled external field, and/or there is a contribution of the magnetic dipole term $\langle T_k \rangle$, since $m_{S_{eff}} = -2\langle S_k \rangle - 7\langle T_k \rangle$, where $\langle S_k \rangle$ is the expectation value of the electron spin projected along the **k** vector of the x-ray beam. Yamasaki et al. discuss the $\langle S_k \rangle$ and $\langle T_k \rangle$ for a single kagome layer case and conclude that the pure AFM spin configuration with positive chirality of the spin (the in/out and chiral AFM condition in Fig. 1 without any canting to the *c*-axis) yields $\langle T_k \rangle = 0$[32]. Since the net $\langle S_k \rangle$ is also 0 for the AFM configuration, we

can assume that the XMCD signal we observe here is not from a pure AFM configuration. Meanwhile, the pure $c$-axis FM condition within the approximation by Yamasaki et al. will lead to a collinear $\langle S_k \rangle$ and $\langle T_k \rangle$, which are hard to be disentangled. The real situation in $Co_3Sn_2S_2$ can be more complex than what they describe.

We further confirm the strong magnetic anisotropy in $Co_3Sn_2S_2$ by showing the magnetic hysteresis loops in Fig. 2d measured at $T = 50$ K by following the XMCD at the $L_3$ peak. The applied magnetic field in Fig. 2b ($B = 1$ T) and Fig. 2c ($B = 6.8$ T) are large enough to saturate the magnetic moment in $Co_3Sn_2S_2$. The saturated XMCD signal (normalized to the $L_3$ peak intensity) in Fig. 2d at a grazing incidence angle of 60° drops to $\frac{2}{3}$ compared to the 0° value agreeing with the sum-rule result. The sudden signal flip at the switching field suggests that the magnetization reversal in $Co_3Sn_2S_2$ occurs by domain wall motion and not by gradual rotation of the moments. Surprisingly, we observe a shift in the hysteresis loop, reminiscent of exchange bias, which typically occurs in an artificial system of layered FM and AFM structures[33], at both incident angles suggesting that $Co_3Sn_2S_2$ does not behave like a normal FM material. Instead, it shows the asymmetry of the magnetic moment along the $c$-axis direction (it needs a larger negative field strength to flip the signal than the positive field) even after saturation is seemingly reached, which may suggest the presence of an AFM phase beside the FM phase at $T = 50$ K. Such coexistence is unusual for a single crystal stoichiometric system. However, there has been a report of exchange bias in self-flux grown $Co_3Sn_2S_2$, which was attributed to a spin glass phase arising from geometric frustration of the kagome lattice[34]. A more recent report[35] on exchange bias in self-flux grown $Co_3Sn_2S_2$ speculates its origin to the existence of AFM at the domain walls. Both report the exchange bias down to low temperatures of 2 and 4.2 K.

## Laser ARPES

The XMCD suggestion of an AFM phase demands further exploration of the system with spatial resolution capability. Up to now, standard spatially resolved magnetic probes such as magnetic force microscopy or magneto-optic Kerr effect techniques have not been successful in detecting AFM regions[36,37]. For this, we utilize instead the spatial resolution of laser $\mu$-ARPES to investigate the fingerprint of the AFM phase at $T = 6$ K. In this work, we report three different cleaved surfaces that we call sample 1, sample 2, and sample 3 as shown in Fig. 3. It should be first noted that most of the area investigated in all three samples do not display sharp dispersing features and instead we observe a broad and smeared band. However, its integrated energy distribution curve (EDC) shows a signal with a typical shape as shown in Fig. 3 with a broad peak around $-0.225\,\mathrm{eV} < E_B < -0.175\,\mathrm{eV}$. In addition, we discover two independent features: a flat band right at $E_F$ and a rather broad band structure that we call the "butterfly" shape. They are independent of each other as each can be seen separately: flat band alone in sample 1, butterfly band alone in sample 3, and both observed together in sample 2. The areas where the flat band and/or the butterfly shape are observed are relatively rare for each cleaved plane as shown in the intensity ratio panel of Fig. 3 (see also Supplementary Fig. 2 for more details). We have cleaved other samples whose ARPES signal is only the majority broad and smeared band, without any flat band at $E_F$ or the butterfly shape. We also note that in general, the synchrotron-based result shows no band like the butterfly within the momentum-energy window covered by the laser ARPES, in any $k_z$ position, as shown in Fig. 1b. Thus, the majority of the area shown by the laser ARPES agrees qualitatively with our synchrotron-based ARPES, which shows no visible band.

We proceed to analyze these findings with DFT calculations and explore the origin of the butterfly shape and the flat band. At first, we notice that this butterfly shape is similar to the case of AFM R-3m and R-3m'calculation[26]. However, the energy of the bands does not match our ARPES data as the calculated band lies at a smaller binding energy (closer to $E_F$). The band energy renormalization that is discussed before for Fig. 1b will not resolve this issue since the bands need to move to a higher binding energy and not closer to $E_F$. We overcome this issue by introducing a larger magnetic moment used in the DFT calculations than $\mu_{Co} \approx 0.33\,\mu_B$. The

value $\mu_{Co} \approx 0.33\,\mu_B$ has been used in several DFT calculations[12,27] as it has been obtained from various bulk measurements[9,11]. However, this small value of cobalt magnetic moment raises a question about its microscopic origin as the cobalt atom magnetic moment can be as high as $\mu_{Co} \approx 2.2$-$2.7\,\mu_B$ for a system with a small orbital contribution[38] and in the PM phase of $Co_3Sn_2S_2$, $\mu_{Co} \approx 1\,\mu_B$[8,11]. Meanwhile, in the case of Co-$TiO_2$, the reported cobalt moment is $\mu_{Co} \approx 0.32\,\mu_B$, which they attribute to the low spin state of the cobalt, i.e., the magnetic moment is coming from the spin (not orbital) of the cobalt[39]. It has been reported that $\mu_{Co} = \frac{1}{3}\mu_B \approx 0.33\,\mu_B$ in the FM case is trivial from a shared cobalt triangle cluster[28]. However, the situation in the in-plane AFM case of $Co_3Sn_2S_2$ may lead to a different effective cobalt moment, which cannot be captured easily with the DFT.

Thus, we explore different DFT results as we vary the value of $\mu_{Co}$ and we settle with $\mu_{Co} \approx 0.6\,\mu_B$ as it matches the observed butterfly band shape and energy position (Fig. 3) that we probe at $k_z \sim 0.6\Gamma Z$ with 6.01 eV (see Supplementary Fig. 3 for details). We notice that for different values of $\mu_{Co}$, the shape of the band remains qualitatively the same and the main effect of varying $\mu_{Co}$ is to shift the position of the bands (See Supplementary Figs. 5–21). Our ARPES data agree better with the pure in-plane chiral AFM phase DFT result, where we only have a degenerate band at the $\bar{\Gamma}$ point (nodal line along the $k_z$ direction) as we can see only one broad peak at the EDC (Fig. 3). However, we cannot rule out the pure in/out AFM phase that predicts split bands at the $\bar{\Gamma}$ point, even though the splitting is not visible in the raw-ARPES result. Meanwhile, we associate the "blurred" band and yet a visible peak on the EDC in the majority area with the FM $\mu_{Co} \approx 0.35\,\mu_B$ calculation (and energy renormalization) by considering the incoherent peak scattered from the majority FM phase. Figure 3 sample 2 shows a clearer FM cut as compared to sample 1 and sample 3 majority regions, which demonstrate that this blurred band has some spatial variation and is also sample dependent.

We attribute the visible flat band near $E_F$ (Fig. 3) to a band in the chiral AFM phase originating from sulphur termination as suggested by our DFT calculation with $\mu_{Co} \approx 0.35\,\mu_B$ (see Fig. 3 and Supplementary Figs. 22–25 for more details). This flat band in the chiral AFM phase on sulphur termination coincides with the experimental finding by STS in a previous report[31]. However, in their case, they attributed the peak found around $E_F$, on both the S-termination (001) surface and the side surface but not on the Sn-termination (001) surface, to come from the bulk FM band at $k_z = 0$ (as also suggested by our DFT calculation in Supplementary Fig. 5). It should be noted that this flat band near $E_F$ is not visible in our synchrotron-based ARPES result, further suggesting that this flat band is a rare occurrence. We quantitatively analyze the flat band by fitting the EDC of the peak with a Lorentzian peak convolved with the detector response (4.7 meV energy resolution) multiplied by the Fermi Dirac distribution. We obtain the peak width, which is resolution limited, to be $\Delta E \sim 3$ meV (half width half maximum) or scattering time of $\tau \sim 100$ fs, which demonstrates that this state is a coherent state and localized on the surface (ideal two-dimensional state) decoupled from the influence of the bulk states. Whether this flat band found in ARPES is the same flat band reported previously[31] is open for discussion. In our case, since the flat band is found independently on both areas with a blurred band and also a butterfly band, it gives strong evidence that it is a surface band rather than a bulk band. This implies the existence of an AFM surface on top of a FM bulk (flat band + blurred band) or an AFM surface on top of an AFM bulk (flat band + butterfly). Meanwhile, since we also find both blurred and butterfly area without the flat band, the AFM surface decoration is shown to be independent from the bulk. The difference in the effective magnetic moment $\mu_{Co}$ for the AFM surface and the majority FM bulk area ($\mu_{Co} \approx 0.35\,\mu_B$) vs the in-plane AFM bulk ($\mu_{Co} \approx 0.6\,\mu_B$) may seem controversial but it is plausible given the peculiar magnetic behavior of $Co_3Sn_2S_2$ and the precise nature of the magnetic ground state not being settled yet. In addition, it is possible for the magnetic moment on the surface to be different from that in the bulk[40–42].

Finally, we demonstrate that the butterfly-shaped band is of magnetic origin by monitoring how the butterfly band at a fixed place (taken from sample 3 in Fig. 3) evolves as a function of temperature towards 200 K where

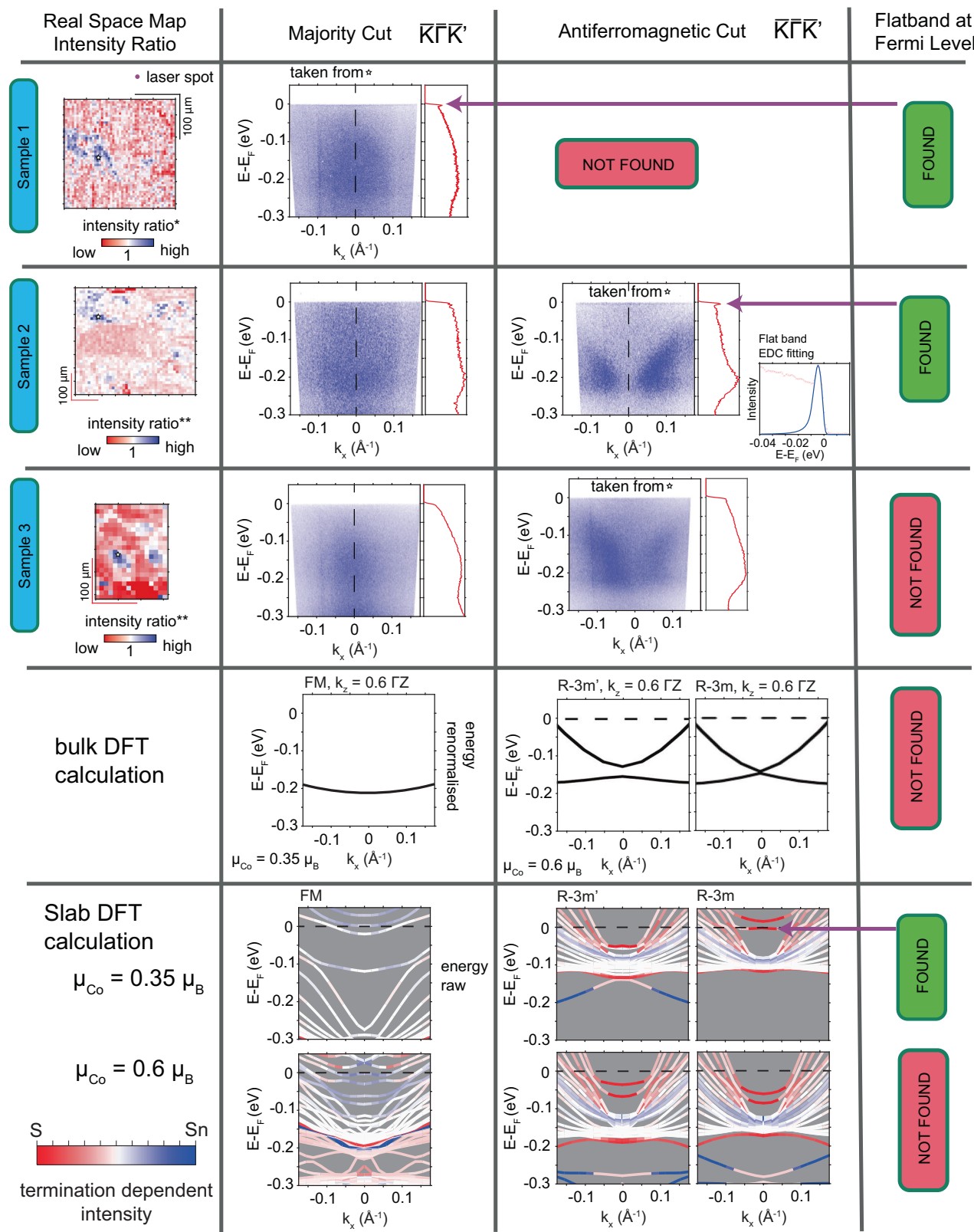

**Fig. 3 | The laser ARPES results for three different samples and their comparison to DFT calculations.** Summary of the real-space map obtained from the intensity ratio * and ** (intensity ratio as explained in Supplementary Fig. 2), example of the $(E, k)$ dispersion along the $\overline{K}\overline{\Gamma}\overline{K}$ of the ferromagnetic band in the majority of the area, example of $(E, k)$ dispersion along the $\overline{K}\overline{\Gamma}\overline{K}$ of the antiferromagnetic band located at blue region of the spatial map, and checklist of flat band at $E_F$ for the three different cleaved samples of $Co_3Sn_2S_2$. The inset shows the flat band EDC fitting in sample 2. The next row is the bulk DFT calculation for the pure ferromagnetic system and the

two possible antiferromagnetic phases of $Co_3Sn_2S_2$ at the $k_z$ position probed by the laser energy (6.01 eV) using effective $\mu = 0.35\,\mu_B$ for the FM phase and $\mu = 0.6\,\mu_B$ for the in-plane AFM phase. The last row is the slab calculation with the color scale indicating the location of the calculated band (from the S termination towards the Sn termination), for both $\mu_{Co} = 0.35\,\mu_B$ and $\mu_{Co} = 0.6\,\mu_B$. The sulphur termination with $\mu_{Co} = 0.35\,\mu_B$ hosts a relatively flat band at $E_F$, which could explain the flat band observed in the ARPES measurement.

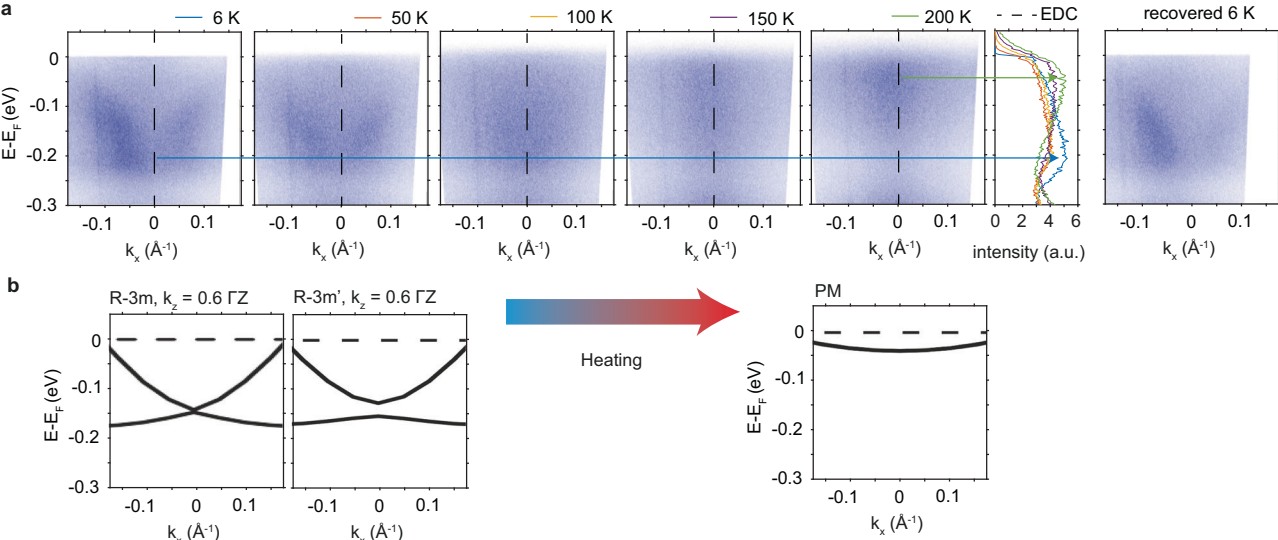

**Fig. 4 | Temperature dependent laser ARPES data showing the change of the band structure. a** Temperature dependent dispersion cut in the antiferromagnetic region showing that the butterfly shape transforms into a broad band close to the Fermi energy $E_F$, whose EDC can be traced to shift up closer to $E_F$ upon warming, agreeing with the paramagnetic band position from the bulk DFT calculation in (**b**). This butterfly shape is reproducible around the same area after re-cooling.

it is known to have transitioned to the PM phase. We can see in Fig. 4 that the band structure gradually changes such that the peak of the EDC moves closer to $E_F$ while the band itself is still broad. This broad band energy position agrees with the PM phase energy position as suggested by the DFT result in the lower panel of Fig. 4, which is close to $E_F$. The origin of this broadness can be due to a trivial $k_z$ broadening as suggested by our slab calculation for the PM phase (see Supplementary Figs. 22–25 for details), but we do not exclude the possibility of magnetic disorder as suggested before[28]. Lastly, we further confirm the magnetic origin of the band by lowering the temperature back to the base temperature where we recover the butterfly shape located around the same area.

The combined findings from XMCD and $\mu$-ARPES suggest that a minority AFM phase exists at low temperature in $Co_3Sn_2S_2$. While this AFM and FM coexistence at low temperature has been reported in doped compounds of $Co_3Sn_{2-x}In_xS_2$ with $x$ ranging from 0.05 to 0.3[18], this can be due to compositional/structural/strain inhomogeneity in a non-stoichiometric compound. Our result suggests the existence of a minority AFM phase in the absence of doping in a stoichiometric compound. Owing to its small fraction compared to the FM phase, it is unlikely to be detected by bulk techniques such as neutron scattering or magnetometry. Among the three different sample growth techniques that have been used, such as self-flux method, Bridgeman technique, and chemical vapor transport (CVT) technique (see Methods for crystal growth technique), the self-flux and CVT method have reported the observation of exchange bias. Other CVT grown $Co_3Sn_2S_2$[43] but with different technical steps exhibits a pure FM hysteresis loop, which could be due to the large field range of the hysteresis loop[44]. The absence of exchange bias in some of the other reported $Co_3Sn_2S_2$ studies may be due to the same reason. The phase coexistence of FM and AFM in $Co_3Sn_2S_2$ may have a similar origin to that observed in stoichiometric pyrochlore $Yb_2Ti_2O_7$[45], with both systems having the ingredients for magnetic frustration giving rise to multiphase competition.

## Conclusions
In summary, we have discovered a new electronic phase in $Co_3Sn_2S_2$ at 6 K associated to a magnetic phase. We associate this new electronic phase to an AFM minority phase in $Co_3Sn_2S_2$, which was previously assumed to be in a pure FM phase. In this work, the AFM phase is first indirectly suggested by the XMCD results showing an exchange bias in the hysteresis loop (a typical indicator of the presence of an AFM material next to a FM material) at a temperature of 50 K, lower than the temperature range of 90–175 K where

AFM has been reported by $\mu$SR. Our spatially resolved laser $\mu$-ARPES in combination with the DFT calculation further confirms that there is an AFM phase co-existing as a minority phase with the majority FM phase at 6 K. As ARPES only probes the first few layers of the compound, it is still an open question whether such pockets of pure AFM phase exist in the bulk of $Co_3Sn_2S_2$. Our results still do not resolve the question of whether the AFM and FM phases compete at higher temperatures or doped compounds as reported by $\mu$SR or it is (mostly) a FM phase with the possibility of canting as suggested by neutron scattering and second harmonic generation experiments. Nevertheless, our work further enriches our understanding of the magnetism of $Co_3Sn_2S_2$, which is still controversial, and suggests an inhomogeneous electronic band structure associated to the existence of a minority AFM phase in the majority FM phase, which is relevant in the context of Weyl semimetals and the interface of topologically different electronic domains. The Weyl points are different dependent on the magnetic phase[26,46] and by having an AFM phase in a sea of FM phase, an interface between two different Weyl semimetals exists, which can give rise to exotic physics[47–50]. This work also invites further theoretical consideration of the spatial co-existence of two different phases in a single stoichiometric compound and further search for other materials that host such co-existence. In addition, we discover a two-dimensional (surface) state in the form of a sharp flat band at the Fermi level, which is decoupled from the short-lived bulk band. Lastly, our finding that the orbital magnetic moment of Co in $Co_3Sn_2S_2$ is negligible should be taken into account when considering physical effects that rely on spin-orbit coupling.

## Methods
### Sample growth and characterisation
Single crystals of $Co_3Sn_2S_2$ were grown by a chemical vapour transport (CVT) method using 2 g of polycrystalline reactant and 160 mg of iodine as a transport agent. The starting components were mixed in helium-filled glovebox, subsequently transferred, cryogenically cooled, evacuated, and sealed in a quartz ampule (ID = 14 mm, OD = 18 mm, and L = 100 mm) under vacuum ($<7\cdot10^{-3}$ mbar). The specimen was placed in a temperature gradient 900- > ~ 800 °C and held over 40 days. Polycrystalline $Co_3Sn_2S_2$ (source) was synthetized from stoichiometric amounts of Co (3N7, Alfa Aesar), Sn (4 N, Alfa Aesar), and S (5 N, Alfa Aesar) by solid state reaction as described elsewhere[17]. Laboratory Powder X-Ray Diffraction (PXRD) measurement performed at room temperature in the Bragg-Brentano geometry using a Bruker AXS D8 Advance diffractometer (Bruker AXS

GmbH, Karlsruhe, Germany) equipped with a Ni-filtered Cu Kα radiation and a 1D LynxEye PSD detector has proven that the obtained crystals are single phase with the Shandite-type structure (Space Group R-3m, No. 166). Small-spot single crystal XRD measurement was also performed to confirm the small volume of the crystal is homogeneous. A crystal of $Co_3Sn_2S_2$, $0.08 \times 0.06 \times 0.02$ mm$^3$, was mounted on the MiTeGen MicroMounts loop and used for x-ray structure determination. Measurements were performed at 80 K and RT (295 K) using the STOE STADIVARI diffractometer equipped with a Dectris EIGER 1 M 2 R CdTe detector and with an Anton Paar Primux 50 Ag/Mo dual-source using Mo Kα radiation (λ = 0.71073 Å) from a micro-focus x-ray source and coupled with an Oxford Instruments Cryostream 800 jet. Data reduction was performed with X-Area package (https://www.stoe.com/products/xarea/). The crystal structure was solved and refined using JANA2020 software[51]. The refined crystal structure is consistent with the reported R-3m one for both 80 K and RT measurements. Refinement reveals no deviations in stoichiometry. In addition, Laue diffraction confirmed high quality of obtained single crystals and EDX measurement confirmed the composition of the compound.

Cleaving possibility is shown in Supplementary Fig. 1. It is most probable to cleave the Sn-S1 and Sn-S2 planes. Therefore, Supplementary Fig. 1(iii) and (iv) configurations are less probable to be found than the (i) and (ii) configurations.

### X-ray magnetic circular dichroism (XMCD) and sum rule

X-ray magnetic circular dichroism (XMCD) was performed in total electron yield (TEY) mode at the X-Treme beamline, Swiss Light Source, Paul Scherrer Institute, Switzerland[29]. The schematic of the measurement is shown in Fig. 2 revealing that the magnetic field applied is always collinear with the direction of the x-ray beam. The sample normal is rotated with respect to the incoming x-ray beam. The x-ray absorption spectra (XAS) are measured by changing the x-ray energy across the cobalt $L_3$ and $L_2$ peaks. Subsequent XMCD is calculated by taking the difference in the absorption spectra between the C+ and C- polarization

$$I_{\text{XMCD}} = I_{\text{C+}} - I_{\text{C-}} \tag{1}$$

where C+ and C- polarization refer to the helicity of the photon being parallel and antiparallel to the photon **k** vector, respectively.

The hysteresis loop is obtained by measuring the TEY signal at a specific energy (peak of $L_3$) and the pre-edge TEY signal in the following order (to minimize drift effect) with magnetic field ramp indicated:
1. Ramp magnetic field to 6.8 T without measurement
2. Ramp to -6.8 T with C+ polarization while taking the TEY signal of the $L_3$ peak and the pre-edge
3. Ramp to +6.8 T with C- and perform measurement as step 2
4. Ramp to -6.8 T with C- and perform measurement as step 2
5. Ramp to +6.8 T with C+ and perform measurement as step 2.

The final XMCD hysteresis signal shown in Fig. 2 is obtained by first taking the difference between the signal at the peak of $L_3$ and the pre-edge for each photon helicity, followed by taking the difference between measurements with C+ and C- for both UP and DOWN ramp, respectively, and normalized by the averaged pre-edge subtracted $L_3$ signal to compensate the different beam footprints between 0 and 60°.

Background subtraction of the XAS signal is performed by fitting the background using the formula

$$f_{\text{bg}} = A_1 \times \left( \frac{2}{3} \arctan\left( a \times \left( E - E_{L_3} \right) \right) + \frac{1}{3} \arctan\left( a \times \left( E - E_{L_2} \right) \right) \right)$$
$$+ A_2 \times E + A_3 \tag{2}$$

$E$ is the photon energy and the fitting variables are
$A_1$: the intensity related to the arctan function

$a$: the slope of the rising background from arctan
$A_2$: the intensity related to a linear function
$A_3$: a constant.

We fix the values $E_{L_3}$ and $E_{L_2}$ as the energy positions of the middle part of the step functions of the background. This is arbitrary as long as they are within the range of the peak. For our case, we choose the middle energy from the energy range of the peak.

Sum rule calculation is performed with the following formulae[52,53]

$$m_l = -\frac{4}{3}\frac{q}{r} n_h = -\langle L_z \rangle \tag{3}$$

$$m_{S_{\text{eff}}} = -\frac{6p - 4q}{r} n_h = -2\langle S_{z,\text{eff}} \rangle \tag{4}$$

where[53]
- $\langle L_z \rangle$ is the ground state expectation value of the orbital angular momentum operator
- $m_l$ is the orbital magnetic moment (opposite sign to the orbital angular momentum)
- $\langle S_{z,\text{eff}} \rangle$ is the ground state expectation value of the effective spin.
- $m_{S_{\text{eff}}}$ is the spin magnetic moment which is opposite to the spin direction
- $n_h$ is the total number of holes
- $r$ is the integrated intensity from C+ and C- of $L_3$ and $L_2$
- $q$ is the integrated intensity from XMCD of $L_3$ and $L_2$
- $p$ is the integrated intensity from XMCD of $L_3$.

In the X-Treme configuration, a negative value for $m_{S_{\text{eff}}}$ and $m_l$ from the above formulae implies that the direction of $m_{S_{\text{eff}}}$ and $m_l$ is antiparallel to the **k** vector of the beam (along the applied $B$ field). Thus, the expectation values of the effective spin $\langle S_{z,\text{eff}} \rangle$ and the angular momentum $\langle L_z \rangle$ point opposite to the $B$ field direction. This is the result shown in Fig. 2.

### Angle resolved photoemission spectroscopy (ARPES)

**Synchrotron-based ARPES.** The synchrotron-based ARPES measurements were performed at the ULTRA endstation, Surface/Interface Spectroscopy (SIS) beamline, Swiss Light Source equipped with a Scienta Omicron DA30L analyzer. The photon dependent measurement was performed with a circular plus (C +) polarization light with energy of 34–150 eV and a total energy resolution of 15 meV. The sample was cleaved in situ at a base pressure lower than $5 \times 10^{-11}$ mbar, and measured at 20 K.

### Micro-focused laser angle resolved photoemission spectroscopy (μ-ARPES)

**Experimental setup and condition.** The μ-ARPES measurements were performed in the laboratory of Prof. Felix Baumberger at the University of Geneva. The photon source we use is a continuous laser from LEOS with an energy of 6.01 eV, coming from its 4$^{\text{th}}$-harmonic generation output. For beam focusing, we use a custom-built lens to focus the beam spot diameter to ∼3 μm. The electron analyzer is an MB-Scientific analyzer equipped with a deflection angle mode to map the dispersion relation while retaining the illuminated area (i.e., no sample rotation is needed to map the Fermi surface). Typical energy and angular resolution are 3 meV / 0.2°. The sample is mounted on a conventional 6-axes ARPES manipulator[54] and the sample position is scanned with an $xyz$ stage of 100 nm resolution and < 1 μm bidirectional reproducibility. The pressure during the measurement was kept at $< 10^{-10}$ mbar. A more detailed explanation can be found in ref. 55. The samples measured were always cleaved at the base temperature of 6 K. The sample spatial drift at subsequent higher temperatures is tracked by using the edges of the sample as a reference. The samples are pre-aligned to the high symmetry cut with low energy electron diffraction (LEED) after cleaving.

**$k_z$ broadening effect**. The perpendicular momentum $k_z$ of the electrons measured by ARPES can be obtained from the expression

$$k_z = \sqrt{\frac{2m_e^*}{\hbar^2}\left(K_{out} + V_o\right) - \frac{2m_e}{\hbar^2}K_{out}\sin^2\phi} \quad (5)$$

where $m_e^*$ is the effective mass of the electron, $K_{out} = h\nu - w - |E_b|$ is the kinetic energy of the electron, $h\nu$ is the photon energy, $|E_b|$ is the binding energy of the electron, $w$ is the work function of the detector, $V_o$ is the inner potential of the material ($V_o = 13$ eV according to our photon dependent data in Supplementary Fig. 3 in agreement with the reported inner potential in the supplementary information of this publication[56]), and $\phi$ is the analyzer slit angle (more details in these refs. [57,58]). The laser photon energy used, 6.01 eV, corresponds to a perpendicular momentum in between $k_z = 0$ and $k_z = \pi$ plane, as shown in Supplementary Fig. 3.

Finally, we can estimate $\delta k_z$ from the experimental line-widths using the following relation:

$$\delta k_z \approx \frac{FWHM_{EDC, experiment}}{\left(\frac{\partial E}{\partial k_z}\right)_{DFT}} \quad (6)$$

In our data, the FWHM of the butterfly shape at $k_x \approx 0.1\,\text{Å}^{-1}$ is $FWHM_{butterfly, k_x \approx 0.1\text{Å}^{-1}} \approx 0.10 - 0.15$ eV. The DFT calculation shows that (see Supplementary Fig. 10)

$$\left(\frac{\partial E}{\partial k_z}\right)_{E_F} \approx 0.25\,\frac{eV}{\Gamma Z} \quad (7)$$

These values give us an estimation of $\delta k_z \approx (0.4 - 0.6)\Gamma Z$, agreeing (by the order) with the estimation from the universal curve.

## Density functional theory calculation

We carry out density functional theory (DFT) calculations to gain insight into the electronic property of $Co_3Sn_2S_2$. We use the projector augmented wave (PAW) approach[59] with the generalized gradient approximation in the parametrization of Perdew, Burke, and Ernzerhof[60] for exchange correlation as implemented in the Vienna Ab initio Simulation Package (VASP)[61,62]. For Co and S, standard potentials are used (Co and S in the VASP distribution), while for Sn, a potential in which d states are treated as valence states is used ($Sn_d$). In most cases, we use an $8 \times 8 \times 8$ **k**-point grid and an energy cutoff of 500 eV. The spin-orbit coupling (SOC) is included, but the +U correction is not included because $Co_3Sn_2S_2$ is an itinerant magnetic system. The FM ordering with Co moments pointing perpendicular to the kagome plane is found to be the most stable as observed computationally.

In order to compare the electronic properties with different Co ordered moment and spin arrangement, we also carry out the constrained magnetism calculations considering various AFM states by setting I CONSTRAINED M = 2. We gradually increase the value of LAMBDA so that the penalty to the total energy $E_p$ becomes sufficiently small. After the electronic ground state is obtained, we carry out non-self-consistent calculations to compute the bulk electronic band structure along high-symmetry lines using the optimized charge density. To compute the slab band structure, we use the WannierTools package[63] with the maximally-localized Wannier functions generated by the Wannier90 code[64]. For finite thickness slabs, there could exist two sets of surface terminations, between Sn and S layers and between S and Co3Sn layers. In the supporting information, we present results of finite-thickness slabs in which 10 unit cells are stacked perpendicular to the kagome plane with Sn and S surfaces as the other set is hard to be cleaved.

## Ethics

This research was conducted following ethical guidelines, ensuring integrity, transparency, and inclusivity. All contributors have been appropriately credited, and no biases influenced the research or authorship. The study does not involve human or animal subjects, and data have been presented accurately and without manipulation.

## Data availability

All data related to this paper are available at a public repository (MARVEL Materials Cloud Archive), with the same title as this paper (https://archive.materialscloud.org).

## Code availability

The code used for data analysis in this study is available from the corresponding author upon reasonable request.

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

## Acknowledgements

We acknowledge Anja Weber at Mesoscopic Systems, Paul Scherrer Institute – ETH Zurich for her help in measuring the EDX SEM of our sample.

S.A.E acknowledges the support from NCCR-MARVEL funded by the Swiss National Science Foundation, the European Union's Horizon 2020 research and innovation programme under the Marie Skłodowska-Curie grant agreement No 701647, and the European Research Council HERO Synergy grant SYG-18 810451. The research by S.O. was supported by the U.S. Department of Energy, Office of Science, Basic Energy Sciences, Materials Sciences and Engineering Division.

## Author contributions

YS conceived the project. LR and GDJ grew the single crystals. IP did the XRD characterization of the crystals. SAE conducted the XMCD measurements and analysis with the help of JD and input from YS. SAE and AT conducted the ARPES measurements with the help of AH and input from YS. SAE did the ARPES data analysis with input from YS. SO did the DFT calculations with discussions with SAE and YS. SAE and YS wrote the manuscript with input from SO, GDJ, and AT. YS directed the project.

## Competing interests

The authors declare no competing interests.
