## [Transparent Peer Review file · Communications Materials]

Inhomogeneity in Electronic Phase and Flat Band in Magnetic Kagome Metal $\text{Co}_3\text{Sn}_2\text{S}_2$

Corresponding Author: Dr Y Soh

This manuscript has been previously reviewed at another journal. This document only contains information relating to versions considered at Communications Materials.

Version 0:

Decision Letter:

Dear Dr Soh,

Your manuscript titled "New Electronic Phase and Flat Band in Magnetic Kagome Metal $\text{Co}_3\text{Sn}_2\text{S}_2$ " has now been seen again by our referees, whose comments appear below. In light of their advice I am delighted to say that we are happy, in principle, to publish a suitably revised version in Communications Materials.

We therefore invite you to edit your manuscript to comply with our journal policies and formatting style in order to maximise the accessibility and therefore the impact of your work.

EDITORIAL REQUESTS

* Your manuscript should comply with our policies and format requirements, detailed in our style and formatting guide (<https://www.nature.com/documents/commsj-phys-style-formatting-guide-accept.pdf>).

* Please edit your manuscript according to the editorial requests in the attached table, and outline revisions made in the right hand column. If you have any questions or concerns about any of our requests, please do not hesitate to contact me. It is important that each request be addressed in order to avoid delays in accepting your manuscript. Please upload the completed table with your manuscript files as a Related Manuscript file.

* The editorial requests table also includes a full list of the files that must be provided upon resubmission. Please upload your files according to this table.

* An updated editorial policy checklist that verifies compliance with all required editorial policies must be completed and uploaded with the revised manuscript. All points on the policy checklist must be addressed; if needed, please revise your manuscript in response to these points. Please note that this form is a dynamic 'smart pdf' and must therefore be downloaded and completed in Adobe Reader. Clicking this link will download a zip file containing the pdf.

OPEN ACCESS

Communications Materials is a fully open access journal. Articles are made freely accessible on publication. For further information about article processing charges, open access funding, and advice and support from Nature Research, please visit <https://www.nature.com/commsmat/open-access>

Please use the following link to submit your revised files:

Link Redacted

We hope to hear from you within two weeks; please let us know if the process may take longer.

Best regards,
Aldo

Dr Aldo Isidori
Senior Editor
Communications Materials

REVIEWERS' COMMENTS:

Reviewer #1 (Remarks to the Author):

In the previous round of submission to Nature Communications, the authors provided detailed responses to all the raised concerns, including the Weyl fermion states, the origin of the flat band, and the influence of experimental conditions. They also incorporated necessary discussions and additional data into the revised manuscript.

In the current revision, the authors have made further improvements in this submission to Communications Materials. They have adopted more refined analytical methods for certain datasets, enhancing the clarity and readability of the manuscript.

Although some challenges remain in this study, such as the clarity of the photoemission spectra, I believe the experimental results are reliable. Furthermore, the discussions on magnetism and band evolution offer valuable insights for the condensed matter physics community. In my opinion, this manuscript is suitable for publication in Communications Materials.

Open Access This Peer Review File is licensed under a Creative Commons Attribution 4.0 International License, which permits use, sharing, adaptation, distribution and reproduction in any medium or format, as long as you give appropriate credit to the original author(s) and the source, provide a link to the Creative Commons license, and indicate if changes were

made.
